# Expression of Specific Alleles of Zinc-Finger Transcription Factors, *HvSAP8* and *HvSAP16*, and Corresponding SNP Markers, Are Associated with Drought Tolerance in Barley Populations

**DOI:** 10.3390/ijms222212156

**Published:** 2021-11-10

**Authors:** Akmaral Baidyussen, Satyvaldy Jatayev, Gulmira Khassanova, Bekzak Amantayev, Grigory Sereda, Sergey Sereda, Narendra K. Gupta, Sunita Gupta, Carly Schramm, Peter Anderson, Colin L. D. Jenkins, Kathleen L. Soole, Peter Langridge, Yuri Shavrukov

**Affiliations:** 1Faculty of Agronomy, S. Seifullin Kazakh AgroTechnical University, Nur-Sultan 010000, Kazakhstan; bai_akmaral@mail.ru (A.B.); satidjo@gmail.com (S.J.); khasanova-gulmira@mail.ru (G.K.); bekzat-abu@mail.ru (B.A.); 2A.F. Khristenko Karaganda Agricultural Experimental Station, Karaganda Region 100435, Kazakhstan; sereda_t@bk.ru (G.S.); sergey.sereda.00@bk.ru (S.S.); 3Department of Plant Physiology, SKN Agriculture University, Jobner 303 329, India; nkgupta69@yahoo.co.in (N.K.G.); sunita.pphy.rari@sknau.ac.in (S.G.); 4College of Science and Engineering, Biological Sciences, Flinders University, Adelaide, SA 5042, Australia; carly.schramm@flinders.edu.au (C.S.); peter.anderson@flinders.edu.au (P.A.); colin.jenkins@flinders.edu.au (C.L.D.J.); kathleen.soole@flinders.edu.au (K.L.S.); 5Wheat Initiative, Julius-Kühn-Institute, 14195 Berlin, Germany; peter.langridge@adelaide.edu.au; 6School of Agriculture, Food and Wine, University of Adelaide, Urrbrae, SA 5005, Australia

**Keywords:** ASQ and Amplifluor markers, barley, drought, gene expression, grain yield and yield components, SNP plant genotyping, shoots number per plant, stress-associated genes, thousand grain weight, zinc-finger transcription factors

## Abstract

Two genes, *HvSAP8* and *HvSAP16*, encoding Zinc-finger proteins, were identified earlier as active in barley plants. Based on bioinformatics and sequencing analysis, six SNPs were found in the promoter regions of *HvSAP8* and one in *HvSAP16*, among parents of two barley segregating populations, Granal × Baisheshek and Natali × Auksiniai-2. ASQ and Amplifluor markers were developed for *HvSAP8* and *HvSAP16*, one SNP in each gene, and in each of two populations, showing simple Mendelian segregation. Plants of F_6_ selected breeding lines and parents were evaluated in a soil-based drought screen, revealing differential expression of *HvSAP8* and *HvSAP16* corresponding with the stress. After almost doubling expression during the early stages of stress, *HvSAP8* returned to pre-stress level or was strongly down-regulated in plants with Granal or Baisheshek genotypes, respectively. For *HvSAP16* under drought conditions, a high expression level was followed by either a return to original levels or strong down-regulation in plants with Natali or Auksiniai-2 genotypes, respectively. Grain yield in the same breeding lines and parents grown under moderate drought was strongly associated with their *HvSAP8* and *HvSAP16* genotypes. Additionally, Granal and Natali genotypes with specific alleles at *HvSAP8* and *HvSAP16* were associated with improved performance under drought via higher 1000 grain weight and more shoots per plant, respectively.

## 1. Introduction

Zinc-finger transcription factors (ZF-TF) with A20/AN1 and AN1/C2H2 domains encode a large super-family of Stress-associated proteins (SAP). Those containing the A20/AN1 domains make up the largest group, while the second sub-family with AN1/C2H2 domains is much smaller. Many SAP proteins are present in each sub-family and they occur in all groups of living organisms, from microbes to plants, humans, and animals. The important role of ZF-TF proteins has been well described in medical research, starting from the classical Tumor necrosis factor α [1] to the involvement of ZF with A20 domains in arthritis [2], and A20/AN1 domain ZF in glioblastoma invasion [3]. In plants, it is widely recognized that *SAP* genes encoding ZF-TFs are associated with reactions to abiotic and biotic stresses. The most studied *SAP* genes in various plant species are strongly responsive to one or multiple stresses, where they are involved in the regulation of downstream gene networks [4,5,6,7].

During the process of evolution, *SAP* genes have diversified greatly and, therefore, their functions in plants can lead to a range of responses [5]. It is also clear that the presence of domains A20/AN1 or AN1/C2H2 results in differential binding to sites in the promoters of downstream genes leading to differences in gene expression and function. The importance of a particular A20 domain was demonstrated using a series of deletions of *LmSAP,* with A20/AN1 domains, from *Lobularia maritima*, the ornamental sweet alyssum plant species, in transgenic tobacco [8]. The full transgene insertion, or with the A20 domain only, improved tolerance of tobacco to salinity and osmotic stresses. In contrast, overexpression of truncated genetic constructs without the A20 domain, or without both domains, resulted in no differences in the abiotic stress tolerance and also, a hormonal imbalance in the transgenic tobacco plants [8]. There is no published information available about any similar studies of the AN1/C2H2 domains, or their comparison with the A20/AN1 domain. Additionally, it has been proposed that ZF-TF with the A20/AN1 domains can play a role in the ubiquitin pathway of protein turnover, while those with the AN1/C2H2 domains are more related to transcriptional regulation [5].

Cultivated barley (*Hordeum vulgare*), a widely grown cereal crop, is genetically similar to both rice (*Oryza sativa*), and *Brachypodium distachyon*, allowing comparative sequence analysis of barley with these species. In total, 17 *HvSAP* barley genes were identified and described in our previous study, with two genes, *HvSAP8* and *HvSAP16*, representing groups with A20/AN1 and AN1/two C2H2 domains, respectively [9]. Under salinity stress, *HvSAP8* (incorrectly classified earlier as *HvSAP5*) was highly up-regulated (4–7 fold) after 7 days of growth in hydroponics with 150 mM NaCl. Transcript levels then returned to normal after 14 days [9]. Other barley genes, *HvSAP6*, *HvSAP11*, *HvSAP12* and *HvSAP15*, showed similar expression profiles, but with variable magnitude and genotype-dependence. In contrast, no published information about the *HvSAP16* gene in barley is available.

In other plant species, particularly monocots, in response to abiotic stresses, a relatively stable and similar trend was found in the expression profile of *SAP8* homologs. In rice, for example, *OsSAP8* expression in shoots of young plants, dehydrated with mannitol, polyethylene glycol (PEG) and air-dried, increased 2–3.5-fold compared to unstressed plants [4,10,11,12]. Similar high levels of expression were reported with PEG-induced dehydration of *FaZnF* (equivalent to *OsSAP8*) in plants of tall fescue (*Festuca arundinacea*) [13] and *SiSAP4* (similar to *OsSAP8*) in foxtail millet (*Setaria italica*) [14]. In soil-grown plants, drought also resulted in similar up-regulated profiles of *ShSAP1* (=*OsSAP8*) in sugarcane (*Saccharum officinarum*) [15] and in *MusaSAP1* (=*HvSAP8*) in banana (*Musa acuminate*) [16]. In all these cases highly expressed homologs of *SAP8* genes showed a similar pattern of up-regulation and subsequent return to normal unstressed levels of expression at later stages of dehydration and drought. Importantly, these reports show similar expression profiles under salinity stress.

In contrast, very little attention has been paid to homologs of the *SAP16* genes. In rice, the expression of *OsSAP16* showed a similar trend of up-regulation in both dehydration and salinity, but to a lesser extent compared to *OsSAP8* under the same conditions [5]. However, the expression of *TaSAP17* (=*OsSAP16*) showed modest up- and down-regulation in bread wheat seedlings under PEG-induced dehydration conditions [17]. Under salinity stress, *TaSAP17* mRNA levels revealed contrasting genotype-dependent expression profiles. Wheat varieties tolerant or sensitive to salinity showed significantly increased or reduced levels of *TaSAP17* expression, respectively, in hydroponic media containing 250 mM NaCl [17].

Due to evolutionary divergence, dicots have varying numbers of *SAP8* homologs, which makes the comparison of expression more complicated. Starting from *AtSAP2*, representing a single homologous gene in *Arabidopsis thaliana*, two and even four similar genes were found in other dicot plant species. Nevertheless, in all these cases, homologs of *SAP8* genes in dicots showed a higher level of expression followed by a return back to non-stressed levels at later stages in air-drying, PEG-induced dehydration, and salt stress conditions. The only case recording no change in expression was in the rapeseed homolog, *BnaA06g02460D* (=*AtSAP2* = *HvSAP8*) [18]. Down-regulation of *SAP8* homologs was reported in cotton (*Gossypium hirsutum*) [19], woody halophyte Kashgar tamarisk (*Tamarix hispida*) [20], as well as in castor bean (*Ricinus communis*) under dehydration [21], and in *Medicago truncatula* with high salinity [22]. For homologs of *SAP16* genes, the expression patterns were similar in the cases described above, including some down-regulation patterns. The summary and critical review of all publications about homologs of studied genes, *SAP8* and *SAP16*, in various monocot and dicot plant species are present in Appendix A.

Grain yield (GY) is the combined result of several important yield components, such as the size and weight of seeds and spikes, seed number per spike, and tiller number. Many genes are known to be involved in the determination of GY and its components, and it is important to determine those with a substantial and clear impact on the improvement of GY. In barley, *HvSAP12* was associated with higher GY under salinity stress, as reported in our previous study [9], while over-expression of the rice transgene *OsSAP5* in plants of *A. thaliana* resulted in a higher 1000 seed weight under heat stress [23].

In bread wheat, a haplotype III of the *TaSAP1-A1* gene (=*OsSAP8*) was strongly associated with a high 1000 grain weight (TGW) [24]. Based on single nucleotide polymorphism (SNP) in the promoter region of *TaSAP1-A1*, molecular markers were developed and successfully used to select targeted genotypes. Significantly higher TGW was associated with specific haplotypes of *TaSAP1-A1* in China [25], Europe [26], and Belarus [27]. Importantly, based on the provided sequence accession number (KC193579) for *TaSAP1-A1* in the NCBI database [24], the protein accession AGU01535 was retrieved and identified as being the closest homolog to polypeptides of OsSAP8 in rice and HvSAP8 in barley.

Other *TaSAP* genes in wheat were also reported to be involved in various GY components and pleiotropic effects. For example, *TaSAP7* from the A and B genomes [28,29], homologs of *OsSAP17* and *HvSAP17*, influenced not only GY and TGW but also plant height [29] and chlorophyll content [28]. Therefore, marker-assisted selection (MAS) for *SAP* genes in wheat and other crops was shown to be successful for practical plant breeding. Various molecular markers are used for MAS, but SNPs are most commonly used for genotyping (Reviewed in: [30]). Amplifluor-like [31] and the recently developed Allele-specific qPCR (ASQ) methods [32] are particularly suitable and non-expensive approaches for plant SNP genotyping, successfully used for this type of research in barley [9,32].

To close gaps in our knowledge on the role of *SAP* genes for drought tolerance and yield production in barley, the aims of the current study were: (1) determine the molecular phylogenetic relatedness of *HvSAP8* and *HvSAP16* genes; (2) characterise the promoter sequences of the genes in the parents of two barley segregating populations; (3) SNP genotype segregating populations for both genes; (4) analyse the gene expression under drought conditions of the parents and selected breeding lines and (5) determine the association between genotype and GY components among parents and selected breeding lines.

## 2. Results

### 2.1. Bioinformatics Analysis and Molecular Phylogeny of HvSAP8 and HvSAP16 Genes

Two genes, *HvSAP8* and *HvSAP16*, examined in this study, represent different sub-types of the same group of Zinc-finger transcription factors. The first gene, *HvSAP8* (MLOC_43986), located on chromosome 7H, represents the most typical group, encoding a Zinc-finger protein (BAK03538 = AK372340) with A20 and AN1 domains. The HvSAP8 polypeptide is very similar to paralog HvSAP4. This is clearly observed in the comparison between the two sub-clades, A1 and A2 (Figure 1a). These sub-clades show perfect discrimination between SAP4 and SAP8 proteins encoded by homologous genes in cereals, including *Brachypodium* as a model species, together with cultivated and wild rice, foxtail millet, maize and sorghum. Strong conservation is observed for SAP8 proteins among other non-cereal monocot plants (Clade B), where African oil and date palms, Malaysian banana, the orchid *Phalaenopsis equestris* and asparagus are clustered together and quite isolated from both sub-clades A1 and A2 in cereal species. The evolutionary divergence of SAP8 between cereal and other monocot plant species is much greater than that between SAP8 and SAP4 within cereals (Figure 1a).

The second gene, *HvSAP16* (MLOC_52196), located on chromosome 2H, represents a relatively rare group encoding a Zinc-finger protein (BAJ92190 = AK360983) with AN1 and two C2H2 domains. The HvSAP16 polypeptide has a very similar distribution among monocot plants (Figure 1b). Cereals show simple clusters, depending on botanical similarity with isolated groups of barley/*Aegilops*/*Brachypodium*, following foxtail millet/sorghum/maize, and wild/cultivated rice. Non-cereal plant species are also isolated and similar to SAP8 proteins (Figure 1).

### 2.2. SNP Identification and Assessment in Parents of Barley Populations

The genes, *HvSAP8* and *HvSAP16*, were identified in reference genomes of varieties Morex, Barke and Bowman, where the sequences of the Open reading frames (ORF) were highly conserved, and hence to find polymorphisms in the barley genotypes used in this study was unlikely. Therefore, promoter regions about 1 Kb upstream from the Start-codon were targeted for sequencing.

For *HvSAP8*, six SNPs were found in the promoter region of barley cv. Granal that differed from the other barley accessions, including the parental form cv. Baisheshek. All six SNPs were identical to those occurring in the Morex reference genome. Two of the SNPs are shown in Figure 2a, and SNP-2 was selected and used in the following genotyping analysis.

Only one SNP was found in the promoter region of *HvSAP16* in barley cv. Auksiniai-2, identical to that in the reference genome of Barke. This SNP differed from the other barley accessions, including the parental cv. Natali (Figure 2b). All SNPs identified had high sequence quality scores, and were verified by repeated sequencing of the genetic region in both directions.

### 2.3. SNP Genotyping of Two Barley Hybrid Populations G×B and N×A Using ASQ and Amplifluor Methods

SNP-ASQ markers for *HvSAP8* and *HvSAP16* were developed and applied for genotyping of corresponding genes in two F_3_ hybrid populations of barley, G×B (Granal × Baisheshek) and N×A (Natali × Auksiniai-2), respectively, with the parents used as references. Genotyping was carried out twice using different instruments and laboratories. The first round of *HvSAP8* allele discrimination in the G×B population and *HvSAP16* alleles in the N×A population was conducted in S.Seifullin Kazakh AgroTechnical University, Nur-Sultan (Kazakhstan), using a Thermo Fisher Scientific QuantStudio-7 Real-Time PCR instrument. An example of a *HvSAP8* genotyping score is presented in Figure 3a. The second round of genotyping, based on the same markers and DNA, was carried out using a Bio-Rad CFX96 Real-Time PCR System instrument at Flinders University, Adelaide (Australia). An example of *HvSAP16* allele scoring is presented in Figure 3b. Both rounds of genotyping showed identical results.

The number of homo- and heterozygote genotypes identified among 42 plants in the G×B progeny and 50 plants in N×A populations, respectively, did not differ from classical Mendelian segregation for *HvSAP8* and *HvSAP16* studied genes. It was verified by Chi-square analysis for each gene (Table 1), where the standard *χ*^2^ value for 1:2:1 type of segregation with df = 2 was *χ*^2^st = 4.61 (*p* < 0.01), indicating the absence of genetic disturbance in progeny genotypes in the two studied hybrid populations. All homozygote plants were used in the further development of breeding lines with known *HvSAP8* and *HvSAP16* genotypes. Based on the developed SNP-ASQ markers, HvSAP8 and HvSAP16, three homozygous breeding lines for each allele were selected in both genes, at generation F_6_, based on a single seed descent strategy, and used in further experiments in this study.

### 2.4. HvSAP8 and HvSAP16 Expression in Drought-Stressed Barley Parents and Breeding Lines Selected from Hybrid Populations

Three selected breeding lines (generation F_6_) from G×B and N×A populations with homozygote alleles of *HvSAP8* and *HvSAP16* genes, respectively, were used, in addition to the parents, for expression analysis in response to mild, moderate and strong drought conditions (Figure 4).

For the *HvSAP8* gene, in the G×B population, all three breeding lines (L8, L30 and L38) and the maternal parent cv. Granal, with the rare ‘*a*’ allele and identical homozygote genotypes showed strong up-regulation (1.7–1.9-fold) at day 8 of mild drought with a gradual return back to the normal level at day 16 of strong drought (Figure 4a). Three other breeding lines (L11, L13 and L18), together with the paternal parent cv. Baisheshek and the ‘*bb*’ genotype of the *HvSAP8* gene, also demonstrated significantly increased expression at day 8 of mild drought, but the level was a little lower (1.3–1.5-fold). However, the following return back to the normal non-stressed level occurred earlier, at day 12 of moderate drought and down-regulation continued at day 16 of strong drought to a level of 0.4–0.5-fold (Figure 4a).

Cross-examination of the same *HvSAP8* gene in selected breeding lines and parents of the N×A population with the non-polymorphic ‘*bb*’ genotype are shown in Figure 4b. Despite some variability, the expression pattern of all genotypes was similar to the Baisheshek haplotype, with 1.3–1.6-fold up-regulation at day 8 of mild drought and down-regulation to 0.2–0.5-fold at day 16 of strong drought stress (Figure 4b).

In contrast, the *HvSAP16* gene was polymorphic in the N×A population, where the maternal parent cv. Natali and three selected breeding lines (L232, L234 and L235) with homozygote ‘*aa*’ genotypes were analysed (Figure 4d). The expressed mRNA level of the gene in ‘*aa*’ genotypes at day 6 of mild drought was relatively high, but the expression of the gene was lowered from 2.3-fold to around 2-fold at day 12 of moderate drought and returned back to the non-stressed level in all four genotypes at day 16 (Figure 4d). For the same group, the expression pattern was very different in three other breeding lines (L1-4, L6-3 and L20-5) with the rare ‘*bb*’ genotype, identical to the paternal parent cv. Auksiniai-2. These four ‘*bb*’ genotypes showed a slight increase in expression, but only two of them were significant at day 8 of mild drought. However, the mRNA levels dropped dramatically in all four genotypes to 0.6–0.3-fold and 0.3–0.1-fold at days 12 and 16, respectively, with moderate and strong drought (Figure 4d).

The expression of the *HvSAP16* gene was also examined in the G×B population. Despite some variability, the expression pattern was similar among all eight genotypes and showed the same trend as in cv. Natali with the ‘*aa*’ genotype (Figure 4c).

To summarize, expression profiles of *HvSAP8* and *HvSAP16* correlated with drought in both parents and selected breeding lines, with contrasting homozygote genotypes in the G×B and N×A populations, respectively.

### 2.5. Barley Breeding Lines Selected for HvSAP8 and HvSAP16 Genotypes: Drought Tolerance and Grain Yield in Dry Environment

The same six F_6_ breeding lines, from each of two hybrid populations, G×B and N×A, with homozygote genotypes for *HvSAP8* and *HvSAP16* genes, respectively, along with the parents, were used to determine grain yield and yield components at harvest in barley plants grown under moderate drought.

Grain yield per plant was significantly higher in groups of plants with the ‘*aa*’ genotypes of either *HvSAP8* or *HvSAP16* in breeding lines and maternal cultivars Granal and Natali, respectively, compared to the corresponding ‘*bb*’ genotypes in breeding lines and paternal cultivars Baisheshek and Auksiniai-2 (Figure 5a,b). However, genetic components for GY were different in the studied barley.

From the many traits used to determine grain yield, only two, the number of shoots per plant (NS/P), and 1000 grain weight (TGW), showed significant differences within the groups of plants. In the G×B population, no difference was found in NS/P, which varied between 2–2.5 shoots per plant among the eight genotypes (Figure 5c). In contrast, a significant difference of either three or two shoots per plant was identified in breeding lines and parents of the N×A population (Figure 5d). This indicated that grain yield improvement in the group of cv. Natali plants grown under moderate drought were associated with the production of significantly more NS/P compared to the group of cv. Auksiniai-2, but not other traits (Figure 5b,d).

A very different situation was found in the group of genotypes with Granal haplotype compared to Baisheshek haplotype for grain yield. The TGW traits showed bigger and heavier grains, with a significantly higher TGW that varied between 63–71 g compared to 49–57 g in plants of the groups Granal and Baisheshek, respectively (Figure 5e). Therefore, the higher grain yield in the Granal haplotype barley group was due to significantly higher TGW. All eight genotypes in the N×A population were similar, at around 61–62 g for TGW (Figure 5f).

## 3. Discussion

The Zinc-finger transcription factor proteins, encoded by the *SAP* genes, are also known as Stress-associated proteins. Indeed, SAP proteins play an important role in plant responses to various stresses, and therefore, *SAP* genes represent part of the molecular network associated with the adaptation of plants to stress. Here we report genetic polymorphisms found in the promoter regions of two genes, *HvSAP8* and *HvSAP16*, with different domains, A20/AN1 and AN1/C2H2, respectively, and the different expression characteristics of these genes under drought.

The nomenclature of newly identified *SAP* genes in plants can be confusing. The current classification of *SAP* genes and their encoded proteins in numerical order follows a traditional process. Many reports describe *SAP* genes based on a comparison with model plant species, such as *Arabidopsis thaliana* for dicots, and rice or *Brachypodium distachyon* for monocots. The situation becomes more complicated with the introduction of Zinc-finger protein (ZFP) classification from human and animal genetics to *SAP* in plants [10] or the independent classification of AN1 proteins [5]. Therefore, careful analysis of the molecular phylogenetic relationships is required for the different *SAP*, *ZFP,* or *AN1* genes in plants, where the standard model for monocot and dicot plant species is encouraged as a priority, as it was used in the presented study.

Based on published data, *SAP* genes are very responsive to abiotic stresses in plants, including *HvSAP* in barley plants under salinity stress [9]. We show here that expression of *HvSAP8* and *HvSAP16* in response to drought stress follows a similar trend, but differences were genotype-dependent and strongly associated with identified SNPs in parents and selected breeding lines from two segregating barley populations, G×B and N×A. These expression profiles were similar to the wheat *TaSAP17* gene, and homologs in rice *OsSAP16* and barley *HvSAP16*, studied under salinity stress [17] (Appendix A).

In the current study, specific alleles of *HvSAP8* and *HvSAP16* showed a strong association with GY in barley parents and selected breeding lines. The results (Figure 5a,c,e) showed a strong association of the Granal haplotype with higher GY and TGW, similar to results for *TaSAP1-A1* in bread wheat, *OsSAP8* in rice and *HvSAP8* in barley. A strong association between haplotype III (based on SNPs) in the promoter region and high TGW as a component of GY in dry field trials, was shown earlier [24,25,26,27].

In contrast, there are no published reports of the strong association between the Natali haplotype of *HvSAP16* with higher GY due to the increased number of productive shoots in each plant as shown here (Figure 5b,d,f). Nevertheless, over-expression of the wheat *TaSAP17* transgene (=*OsSAP16* and =*HvSAP16*) in *Arabidopsis* showed improved germination and seedling growth in media in the presence of 150–250 mM NaCl [17]. Despite some differences, a similar trend has been found in the present study of *HvSAP16* in barley plants compared with the over-expression of wheat *TaSAP17* under drought stress.

We can conclude that ZF-TF genes, *HvSAP8* and *HvSAP16*, with different A20/AN1 and AN1/C2H2 domains, show similar trends of expression profiles in barley plants grown under drought conditions. However, SNPs identified in upstream regions of the genes were significantly correlated with changes in their expression profiles. Importantly, these changes in expression appear to be associated with improvement in GY in barley plants, although the mechanisms of gene action seem to be quite different. The Granal haplotype at *HvSAP8* showed higher TGW, while the Natali haplotype at *HvSAP16* resulted in increased shoot numbers per plant. Therefore, these SNPs provide a useful tool to assist cereal breeding for drought and other abiotic stress-tolerant cultivars based on MAS.

## 4. Materials and Methods

### 4.1. Bioinformatic Analysis

The full-length nucleotide sequence of *HvSAP* and its corresponding polypeptide sequence were identified using both BLASTN and BLASP in NCBI and from the IPK Barley BLAST Server (https://webblast.ipk-gatersleben.de/barley_ibsc/viroblast.php) (accessed on 16 July 2021). Sequences of homologous genes and encoded proteins in other plants species were retrieved via the KEGG database: Kyoto Encyclopedia of Genes and Genomes (https://www.genome.jp/tools/blast) (accessed on 1 October 2021).

The molecular dendrograms of amino acid sequences in SAP polypeptides from barley and other monocot plants were constructed using the SplitsTree4 program (http://www.splitstree.org) (accessed on 18 June 2021). The algorithm of Unrooted Consensus tree and Equal angle dendrogram option was used for the preparation of both *SAP* genes and *SAP* polypeptide phylogenetic trees.

### 4.2. Plant Material

The four spring barley cultivars with different origins used in this study were developed for animal feed production and described in Table 2.

Hybrid populations G×B and N×A originated from manual emasculation, isolation and controlled pollination of parental plants conducted by Grigory Sereda (A.F.Khristenko Karaganda Agricultural Experimental Station, Karaganda region, Kazakhstan) and Raisa Okovitaya (A.I.Barayev Research and Production Centre of Grain Farming, Shortandy, Kazakhstan), respectively. Progeny F_3_ were produced by self-pollination of F_2_ plants and comprised 42 and 50 offspring in G×B and N×A hybrid populations, respectively. Each of the F_3_ individuals was used as a genetic source of the corresponding breeding line down to the F_6_ generation using self-pollination and single-seed descent strategy. Only six F_6_ breeding lines were selected for this study with contrasting genotypic scores from each of the two populations. A small number of seeds can be provided to researchers upon request, subject to a Material transfer agreement.

### 4.3. Drought Stress Treatment

Barley seeds were germinated in Petri dishes and seedlings were transplanted to 12 L pots, with five uniform seedlings per pot, filled with equal volumes of commercial soil potting mix (Nesterovskoe, Kazakhstan) and clay soil from a nearby research field, as described earlier [36], with at least twice the number of plants than required. Plants were grown for one month in rain-excluding, clear-plastic shaded open space in the Central campus of Agronomy Faculty, S.Seifullin Kazakh AgroTechnical University, Nur-Sultan (Kazakhstan). The conditions of barley growth and time schedule were designed to simulate barley growth in real fields in the region.

For the drought experiment, a well-watered regime was maintained for one month in all pots. The pots were then split into two groups, Controls (continued watering once every two days) and slowly developing drought, where watering was withdrawn. Plants were arranged to collect leaf samples at four consecutive time-points. ‘day 0′ was immediately before the drought treatment started. One plant from the same pot and genotype was used at days 8, 12 and 16, for three following sampling time-points, corresponding to mild, moderate and strong drought, respectively.

Volumetric water content (VWC) in the soil was measured using a portable Moisture meter (Model CS616, Campbell Scientific, Australia). At the sampling time-points, VWC value was decreased in the drought treatment from 80% field capacity (day 0) to 40%, mild drought with symptoms of leaf wilting just starting (day 8). At day 12 with 30% field capacity, moderate drought and clear leaf wilting were observed. Finally, at day 16, plants were severely affected by strong drought, where all leaves were wilted and yellowed, with a corresponding 20% soil moisture of field capacity. Three pots with five plants per pot were grown for each parent and breeding line, and three plants (biological replicates) were used for leaf sampling in each genotype, treatment and time-point.

For agronomic performance, five plants per pot and three pots per genotype of parents and breeding lines were arranged and grown as described above. However, moderate drought was applied for all pots with one-month-old plants, monitoring soil water content to around 30% moisture of field capacity, and then supported with the addition of a small volume of water weekly until plant harvest. The placement of all pots with plants in all experiments within the open space was fully randomized.

### 4.4. DNA Extraction, Sequencing and SNP Identification

Barley plants were grown in control (non-stressed) conditions in pots with soil as described above. Individual one-month-old plants were selected from parents, F_3_ progenies and F_6_ breeding lines from two hybrid populations. One young leaf was collected separately from each plant without bulking of samples. Leaf samples were frozen in liquid nitrogen and ground in 10-mL tubes with two 9-mm stainless ball bearings using a Vortex mixer.

DNA was extracted from crushed leaf samples with phenol-chloroform, as described in our earlier papers [37,38]. One µL of extracted DNA was checked on a 0.8% agarose gel to assess quality, and concentration was measured by a Nano-Drop spectrophotometer (Thermo Fisher Scientific, Waltham, MA, USA). For sequencing, primers were designed flanking a ~1 Kb fragment in the promoter regions of the target genes, *HvSAP8* and *HvSAP16*, based on annotated sequence accessions of Morex, Barke and Bowman in databases. Sequences of the promoter regions with highlighted primers are presented in Appendix A.

Amplification and purification of PCR products were carried out as described earlier [39]. In brief, PCR was performed in 60 µL volume reactions containing 8 mL of template DNA adjusted to 20 ng/mL, and with the following components in their final concentrations as listed: 1×PCR Buffer, 2.0 mM MgCl_2_, 0.2 mM each of dNTPs, 0.25 mM of each primer and 4.0 units of Taq-DNA polymerase (Go-Taq, Promega, Madison, WI, USA) in each reaction. PCR was conducted on a Thermal iCycler (Bio-Rad, Hercules, CA, USA), using a program recommended by the Taq-polymerase manufacturer, with the following steps: initial denaturation, 94 °C, 4 min; 35 cycles of 94 °C for 20 s, 55 °C for 20 s, 72 °C for 1 min, and final extension, 72 °C for 5 min. Single bands of the expected size were confirmed after visualization of 5 µL of the PCR product in a 1% agarose gel. The remaining PCR product (55 µL) was purified using FavorPrep PCR Purification kit (Favorgene Biotec., Taiwan) following the manufacturer’s protocol.

The concentrations of purified PCR products were measured using NanoDrop (Thermo Fisher Scientific, USA) and later used as the template (100 ng) in the reaction with the BigDye Terminator Cycle Sequencing Kit, v3.1 (Applied Biosystems, Thermo Fisher Scientific, Waltham, MA, USA) following the manufacturer’s protocol. Research service for Sanger sequencing using an AB3730xl instrument was provided by AGRF (Australian Genome Research Facility) at Adelaide, Australia. 

SNP were identified using direct comparison of sequence chromatograms of different genotypes and with annotated sequences using the Chromas software program, v. 2.6.2 (www.technelysium.com.au) (accessed on 20 May 2020).

### 4.5. Allele-Specific qPCR (ASQ) and Amplifluor-Like SNP Genotyping

Allele-specific qPCR for SNP analysis was carried out using both QuantStudio-7 Real-Time PCR instrument (Thermo Fisher Scientific, Waltham, MA, USA) and a CFX96 Real-Time PCR Detection System (Bio-Rad, Hercules, CA, USA) using DNA samples as described previously [32] with the following adjustments. Each reaction with a total volume of 10 µL contained: 2 µL of template DNA adjusted to 10 ng/µL, 1 µL of the mix including two FAM- and HEX-fluorescently-labelled Universal probes (3 μM each) and one Universal probe with BHQ1 quencher (6 µM), 1 µL of allele-specific primer mix (1 μM of each of two forward primers and 5 μM of the common reverse primer), and 2 µL of 5×Go-Taq Master-mix (Promega, Madison, WI, USA), with the following final concentration of components per µL of the reaction: 3 mM MgCl_2_, 0.2 mM of dNTP and 0.04 units of Go-Taq polymerase (Promega, Madison, WI, USA).

The annotated SNP sites were used to design allele-specific primers, and the arrangement of HvSAP8 primers is presented in Figure 6 while those for HvSAP16 were published in our earlier paper [32]. All sequences of the Universal probes and primers and sizes of amplicons generated are presented in Appendix A.

PCR was conducted using a program adjusted from those published earlier [32]: initial denaturation, 94 °C, 2 min; 10 initial cycles of 94 °C for 10 s, 55 °C for 20 s, and 68 °C for 20 s; 30 secondary cycles of 94 °C for 10 s, 62 °C for 20 s, 68 °C for 20 s, and 55 °C for 30 s; with the recording of allele-specific fluorescence after each secondary cycle. Genotyping by SNP calling was determined automatically by the corresponding instrument software, but each SNP result was also checked manually using amplification curves and final allele discrimination. The experiments were repeated twice with two technical replicates, confirming the confidence of SNP calls. To verify confidence of allele discrimination of the studied genes, *HvSAP8* and *HvSAP16*, SNP genotyping was repeated once more using an Amplifluor-like method, testing the same DNA samples and following the original protocol [31] with recent modification [40].

### 4.6. RNA Extraction, cDNA Synthesis, and qPCR Analysis

Three individual plants from each parent and from each of the 12 F_6_ breeding lines were selected from the two barley populations (G×B and N×A), grown in pots with soil in Control (well-watered) and drought conditions, as described in Section 4.3 above. One fully developed leaf was collected from each plant under the control and drought conditions at four time-points, days 0, 8, 12 and 16 after water was withdrawn in drought-affected pots. Frozen leaf samples were ground in 10-mL tubes with two 9-mm stainless steel ball bearings using a Vortex mixer, as described in Section 4.4 above for DNA extraction. TRIzol-like reagent was used for RNA extraction, following the protocol described earlier [41,42] with a subsequent RNA quality check via electrophoresis of 1 μL of each RNA sample on a 1.5% agarose gel and quantification of RNA on a NanoDrop (Thermo Fisher Scientific, Waltham, MA, USA). All cDNAs were synthesised from 2 μg of each RNA sample that passed quality control, after 1 μL of DNase treatment (Qiagen, Hilden, Germany) with 15 min incubation at room temperature (22 °C), and the use of SuperScript III Reverse Transcriptase kit (Invitrogen, Waltham, MA, USA) following the manufacturer’s instructions. The quality of all cDNA samples was confirmed by PCR with products generated of the expected size.

Samples of cDNA diluted with water (1:5) were used for qPCR analyses with a QuantStudio-7 Real-Time PCR instrument (Thermo Fisher Scientific, USA) following the qPCR protocol described earlier [39]. The total volume (10 μL) of qPCR in each well included 5 μL of 2×Biomaster HS-qPCR SYBR Blue (Biolabmix, Novosibirsk, Russia), 4 μL of diluted cDNA, and 1 μL of two gene-specific primers (3 μM of each primer) (Appendix A) as per the manufacturer’s recommendation. The stability, accuracy and reproducibility were consistently high in our experiments using a 10 µL volume of qPCR. Thermal cycling conditions included a brief initial melt at 95°C for 3 min, followed by 40 cycles of 95 °C for 5 s and 60 °C for 20 s, and finished with a melt curve from 60 °C to 95 °C increasing by 0.5 °C increments every 5 s. The efficiencies of all qPCR primers were calculated based on the slope of the corresponding calibration line, and their suitability was confirmed. Specificities of target and Reference genes were verified with single distinct peaks on a melting curve and a single band of the expected size in 2% agarose gel electrophoresis. Expression data for the target genes were calculated with normalisation of gene expression relative to the average expression of the two reference genes: *HvADP*, ADP-ribosylation factor 1-like protein (AJ508228), and *HvGAPD*, Glycolytic glyceraldehyde-3-phosphate dehydrogenase (X60343) [43], and their sequences with amplicon sizes are also presented in Appendix A. At least three biological and two technical replicates were used in each qPCR experiment.

The Relative standard dilution method was used in the study based on the ABI Guide for relative quantitation of gene expression using real-time quantitative PCR (http://www3.appliedbiosystems.com/cms/groups/mcb_support/documents/generaldocuments/cms_042380.pdf) (accessed on 1 January 2008), where serial dilutions were applied for each target and reference gene individually. Threshold cycle values were determined based on linear calibration of template cDNA dilution factor and Cq value.

### 4.7. Agronomy Traits: Grain Yield and Their Components

At the harvesting stage, 15 plants of the parents and each of 12 F_6_ breeding lines, after growth in pots with moderate drought conditions, were dried completely in a post-harvest facility and studied for their agronomic traits. From various measured traits for individual plants, only three showed significant differences and are presented in this study: Grain yield (GY) as the total weight of all seeds from separate plants; Number of shoots per plant (NS/P), counted at harvest in each plant and 1000 grain weight (TGW) as manually counted one thousand seeds weighed using a laboratory scale with 0.1 g accuracy. Statistical treatment of the results, described below, was based on the number of biological replicates, *n* = 15.

### 4.8. Statistical Treatment

IBM SPSS statistical software was used to calculate and analyze means and standard errors using ANOVA, and to estimate the probabilities for significance using Student’s *t*-test via IBM SPSS, Statistics Desktop 25.0.0.0. Chi-square treatment was carried out according to [44].

## Figures and Tables

**Figure 1 ijms-22-12156-f001:**
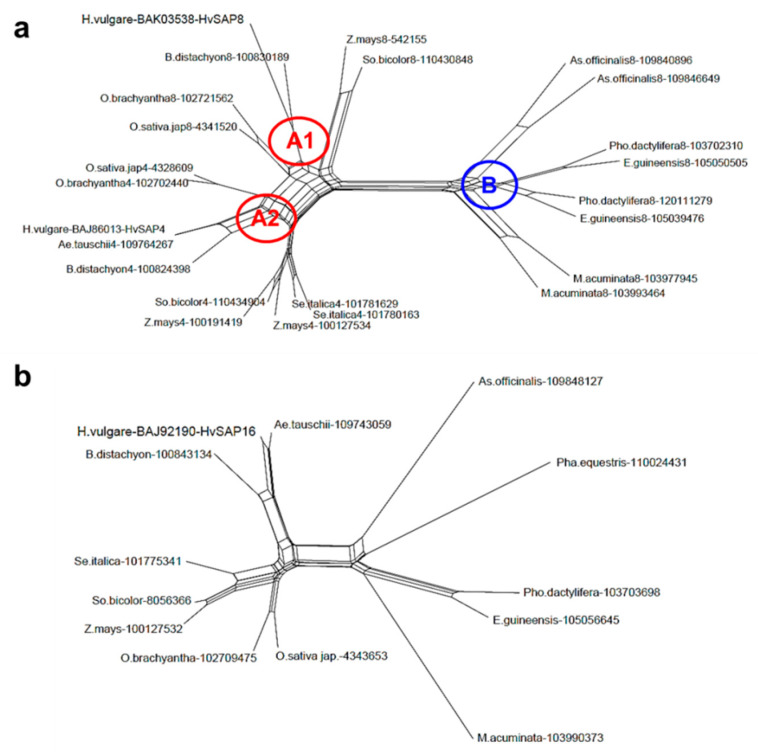
Molecular phylogenetic dendrogram based on amino acid sequences of proteins SAP8 and SAP4 (**a**), and SAP16 (**b**) among monocot species. Sub-clades A1 and A2 designate SAP8 and SAP4 peptides among cereals, and Clade B shows other monocots. Barley HvSAP protein accession IDs were retrieved from the NCBI database and all homologous protein sequences from other monocot plant species were identified in KEGG (Kyoto Encyclopedia of Genes and Genomes). The botanical names of the presented species are as follows, in anti-clockwise order as in the panel (**a**) with the addition of panel (**b**): Clade A: *Sorghum bicolor*, *Zea mays, Brachypodium distachyon*, *Hordeum vulgare*, *Oryza brachyantha*, *Oryza sativa* ssp. *japonica*, *Aegilops tauschii*, and *Setaria italica*; Clade B: *Musa acuminata*, *Elaeis guineensis*, *Phoenix dactylifera*, *Phalaenopsis equestris*, and *Asparagus officinalis*. The corresponding protein sequence accession IDs were added after the species names. The unrooted Consensus tree with Equal angle dendrogram was generated by the program SplitsTree4 (http://www.splitstree.org) (accessed on 18 June 2021).

**Figure 2 ijms-22-12156-f002:**
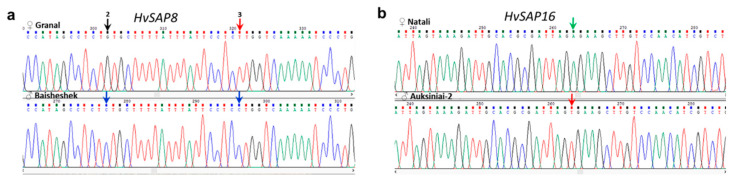
Comparison of sequence fragments of promoter regions of *HvSAP8* (**a**) and *HvSAP16* (**b**) in parents of barley hybrid populations, cultivars Granal, Baisheshes, Natali and Auksiniai-2, respectively. SNPs are indicated by arrows with corresponding colour. Order numbers were used for two SNPs in *HvSAP8* while only a single SNP in *HvSAP16* was found.

**Figure 3 ijms-22-12156-f003:**
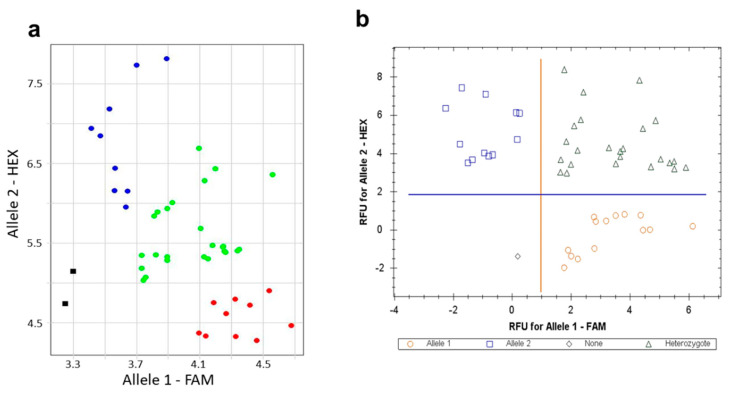
Allele discrimination scores based on SNPs in *HvSAP8* for genotyping of the hybrid population Granal × Baisheshek (**a**) and in *HvSAP16* for genotyping of the hybrid population Natali × Auksiniai-2 (**b**). Relative fluorescence units (RFU) for fluorophores FAM and HEX were transformed automatically into genotyping of alleles 1 and 2, respectively, using a Thermo Fisher Scientific QuantStudio-7 Real-Time PCR instrument (**a**) and a Bio-Rad CFX96 Real-Time PCR instrument (**b**). Homozygotes for allele 1 (FAM) are designated by red dots and red circles; homozygotes for allele 2 (HEX) are shown in blue dots and blue squares; green dots and green triangles represent heterozygotes. Numbers of studied plants were *n* = 42 and *n* = 50, for Granal × Baisheshek and Natali × Auksiniai-2 hybrid populations, respectively. The normalisation was made by the qPCR software with the comparison of fluorescence data to No-template control (NTC, sterile water) shown as black squares in the Thermo Fisher Scientific instrument only.

**Figure 4 ijms-22-12156-f004:**
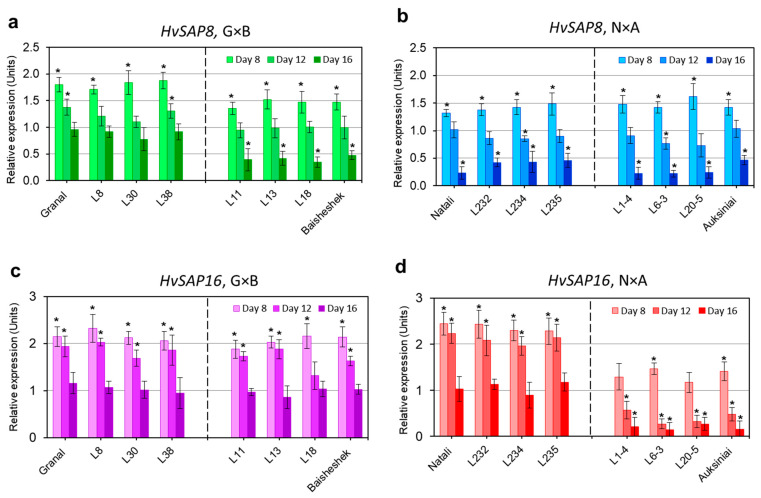
Expression analysis of *HvSAP8* (**a**,**b**) and *HvSAP16* (**c**,**d**) genes in two hybrid populations of barley, Granal × Baisheshek (G×B) (**a**,**c**), and in Natali × Auksiniai-2 (N×A) (**b**,**d**). Each experiment contains eight genotypes including both parents and six breeding lines of each hybrid population. Three breeding lines were selected by SNP genotypes identical to each parent and are separated by a dashed line in the figure panels. Leaf samples were collected from soil-grown plants at four consecutive time-points, where Relative expression at ‘day 0′ (well-watered Controls) was set as unit level 1 for all genotypes and experiments. Three collection time-points were at day 8, 12 and 16, from when water was withdrawn, corresponding to mild, moderate and strong drought treatment, respectively. Expression data were normalised using two Reference genes, *HvADP* (ADP-ribosylation factor 1-like protein) and *HvGAPD* (Glycolytic glyceraldehyde-3-phosphate dehydrogenase), and are presented as the average ± SE of three biological replicates (individual plants) and two technical repeats for each genotype and treatment. Significant differences (*p* < 0.05) from relative level 1 are designated by asterisks (*) above the bars, for each genotype and experiment, calculated using ANOVA.

**Figure 5 ijms-22-12156-f005:**
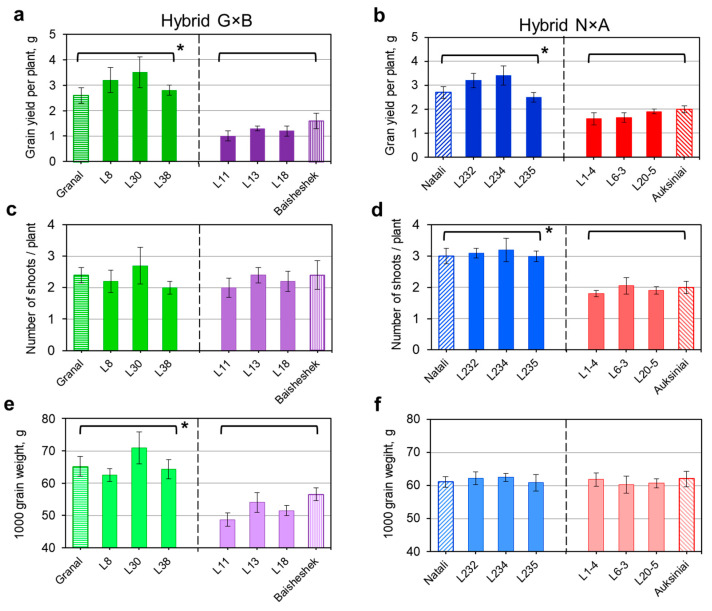
Grain yield and components in barley plants from two hybrid populations, Granal × Baisheshek (G×B) (**a**,**c**,**e**), and Natali × Auksiniai-2 (N×A) (**b**,**d**,**f**), grown in pots with soil under moderate drought until harvest. Eight genotypes including both parents and six breeding lines for the hybrid population identified in this study are separated by a dashed line in the figure panels. Grain yield per plant (**a**,**b**), number of shoots per plant (**c**,**d**), and 1000 grain weight (**e**,**f**), are presented as mean bars ± SE for 15 harvested plants (5 plants × 3 pots) for each genotype. Significant differences (*p* < 0.05) are indicated by asterisks (*) between average values for corresponding groups of genotypes, calculated using ANOVA.

**Figure 6 ijms-22-12156-f006:**
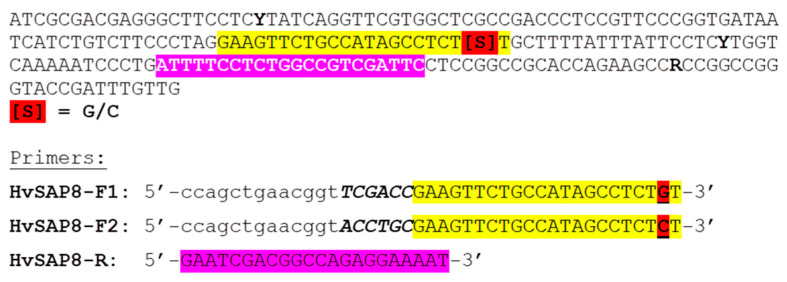
ASQ primer design in the fragment of *HvSAP8* gene promoter sequence in barley parents, Granal and Baisheshek. The targeted SNP is designated by red and other SNP are shown by degenerative nucleotides in the sequence in Bold. Two Forward allele-specific primers (F1 and F2) have a common part indicated in yellow but differ by nucleotides at the SNP position underlined and highlighted in red. At the 5′-end, the 13-bp fragment is common and indicated in lower case letters, while the 6-bp fragment in the middle of both Forward primers is a specific ‘tag’ shown in Bold-Italic case. The common reverse primer is highlighted in pink, with ‘reverse-complement’ order indicated by black and white fonts of letters in Reverse primer and in the sequence, respectively.

**Table 1 ijms-22-12156-t001:** Chi-square analysis for genotype segregations with SNP-ASQ markers HvSAP8 and HvSAP16 in two barley hybrid populations.

SNP Marker and Hybrid	Homozygote *aa*	Heterozygote *ab*	Homozygote *bb*	Total	*χ*^2^ (1:2:1, df = 2)
**HvSAP8, G×B**					
*Observed*	10	23	9	42	
*Expected*	10.5	21	10.5	42	0.43
**HvSAP16, N×A**					
*Observed*	14	24	12	50	
*Expected*	12.5	25	12.5	50	0.24

**Table 2 ijms-22-12156-t002:** Description of four spring barley cultivars used as plant material in this study.

Name	Origin	Accession and Genebank ^1^	Role in the Study	Brief Description	Reference
Granal	Kazakhstan	EBDB ID: 36487, BCC1469	♀, population G×B	Elite grain quality and tolerance to drought	[33]
Baisheshek	Kazakhstan	EBDB ID: 20443, IR3303	♂, population G×B	Tolerance to salinity	[9]
Natali	Russia	K-30957	♀, population N×A	High-yielding, large grains, early ripening	[34]
Auksiniai-2	Lithuania	EBDB ID: 36534, JIC4863	♂, population N×A	Used for mutant production	[35]

^1^—EBDB, European Barley Database, Gatersleben, Germany (https://ebdb.ipk-gatersleben.de/apps/ebdb) (accessed on 14 September 2021); K, the Vavilov Research Institute of Plant Industry, St.-Petersburg, Russia (http://vir.nw.ru/index.html) (accessed on 10 December 2020).

## Data Availability

Data produced in this study is presented in the paper and in the Appendix A.

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
