# Peer review of "Expression of Specific Alleles of Zinc-Finger Transcription Factors, HvSAP8 and HvSAP16, and Corresponding SNP Markers, Are Associated with Drought Tolerance in Barley Populations"

_ijms, 2021, doi:10.3390/ijms222212156_

Round 1

Reviewer 1 Report

The manuscript is well written and the experiments design with great precision.

Only two remarks:

Introduction

It seems inappropriate to include Supply Material link into this section

Material and Methods:

Please put the information about plant material into a table – it will make reading more affordable.

Author Response

Reviewer 1:

The manuscript is well written and the experiments design with great precision.

Only two remarks:

Introduction

It seems inappropriate to include Supply Material link into this section.

Response:

We agree with the Reviewer that, in general, references to Supplementary Material rarely used in Introduction section. In this regard, we deleted such references in L81 and L101, indicated in yellow. However, in L114-116, we inserted our explanation in yellow that the Supplementary Material S1 in fact is a summary of publications (indicated also in L581-583). Therefore, we believe that in this modified form, our reference to the Supplementary material S1 is suitable for the Introduction section.

Material and Methods:

Please put the information about plant material into a table – it will make reading more affordable.

Response:

The sub-section 4.2 has been modified accordingly, including L402 and new Table 2 (L414-416), as suggested by the Reviewer and indicated in yellow. We also swapped references 34 and 35 in Reference list (L692-696) in accordance to the new Table 2. Reference 32 (L689) was adjusted with a DOI identification.

Reviewer 2 Report

Authors have depicted the potential connection between Zinc-Finger transcription factors and corresponding SNP markers associated with Drought Tolerance in Barley. The manuscript was relatively well-performed, and the result supported the goal of the study. Hence, recommended for accepting in present form for publication.

Author Response

Reviewer 2:

Authors have depicted the potential connection between Zinc-Finger transcription factors and corresponding SNP markers associated with Drought Tolerance in Barley. The manuscript was relatively well-performed, and the result supported the goal of the study. Hence, recommended for accepting in present form for publication.

Response:

We appreciate the Reviewer’s comment.